# Gated Attention for Large Language Models: Non-linearity, Sparsity, and Attention-Sink-Free

**Zihan Qiu**[*1], **Zekun Wang**[*1], **Bo Zheng**[*1], **Zeyu Huang**[*2],
**Kaiyue Wen**[3], **Songlin Yang**[4], **Rui Men**[1], **Le Yu**[1], **Fei Huang**[1], **Suozhi Huang**[5],
**Dayiheng Liu**[✉1], **Jingren Zhou**[1], **Junyang Lin**[✉1]
[1]Qwen Team, Alibaba Group [2]University of Edinburgh [3]Stanford University
[4]MIT [5]Tsinghua University

## Abstract

Gating mechanisms have been widely utilized, from early models like LSTMs [1] and Highway Networks [2] to recent state space models [3], linear attention [4], and also softmax attention [5, 6]. Yet, existing literature rarely examines the specific effects of gating. In this work, we conduct comprehensive experiments to systematically investigate gating-augmented softmax attention variants. Specifically, we perform a comprehensive comparison over 30 variants of 15B Mixture-of-Experts (MoE) models and 1.7B dense models trained on a 3.5 trillion token dataset. Our central finding is that a simple modification—*applying an head-specific sigmoid gate after the Scaled Dot-Product Attention (SDPA)—consistently improves performance.* This modification also enhances training stability, tolerates larger learning rates, and improves scaling properties. By comparing various gating positions and computational variants, we attribute this effectiveness to two key factors: (1) introducing non-linearity upon the low-rank mapping in the softmax attention, and (2) applying query-dependent sparse gating scores to modulate the SDPA output. Notably, we find this sparse gating mechanism mitigates 'massive activation' [7], 'attention sink' [8], and enhances long-context extrapolation performance, and we also release related codes and models to facilitate future research. Furthermore, the most effective SDPA output gating is used in the Qwen3-Next models.

## 1 Introduction

Gating mechanism is well-established in neural networks. Early architectures, such as LSTMs [1], Highway Networks [2] and GRUs [9], pioneer the use of gating to control information flow across time steps or layers and improve gradient propagation. This principle persists in modern architectures. Recent sequence modeling works, including state-space models [3, 10] and attention mechanisms [11, 12, 13, 4, 14, 15, 16, 17, 18, 5, 6] commonly apply gating, often to modulate the outputs of token-mixer components. Despite its widespread adoption and empirical success, most recent works do not look into the gating mechanisms like the gating scores and their effect on the model's hidden states.

Insufficient understanding hinders assessing gating's true contribution, especially when confounded with other architectural factors. For instance, while Switch Heads [19, 20] introduces a sigmoid gating to select top-K attention head experts, our experiments reveal an interesting finding (Appendix A.1): substantial performance gains persist even when reduced to a single expert, where the gate simply modulates the value output. This strongly suggests the gating itself provides significant intrinsic value, separate from the routing mechanism. Similarly, in Native Sparse Attention (NSA) [21], while overall performance improvements are demonstrated, they do not disentangle the contributions of its gating mechanism from the effects of the sparse attention design itself. These considerations underscore the need to rigorously disentangle the effects of gating from other architectural components.

39th Conference on Neural Information Processing Systems (NeurIPS 2025).

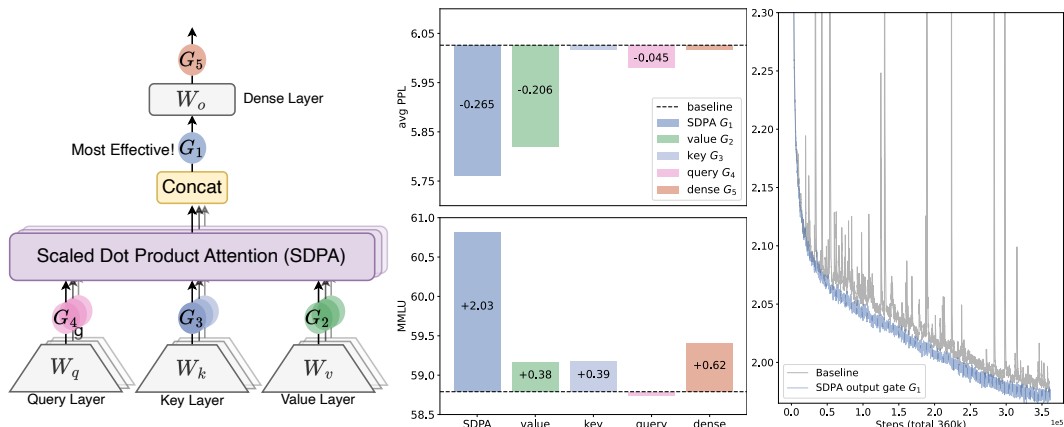

Figure 1: **Left**: Investigated positions for applying gating operations within the self-attention layer.; **Middle**: Performance comparison (Test PPL and MMLU) of 15B MoE models with gating applied at various positions. Gating after SDPA ($G_1$) yields the best overall results. Gating after the Value layer ($G_2$) also demonstrates notable improvements, particularly in PPL. **Right**: Training loss comparison (smoothed, 0.9 coeff.) over 3T tokens between baseline and SDPA-gated 1.7B dense models under identical hyperparameters. Gating results in lower final loss and substantially enhanced training stability, mitigating loss spikes. This stability allows for potentially higher learning rates and facilitates better scaling.

In this work, we investigate gating mechanisms in the standard softmax attention [22] (Sec.2.2). Specifically, we introduce gating at distinct positions (Fig. 1): after the query ($G_4$), key ($G_3$), and value projections ($G_2$); following the Scaled Dot Product Attention (SDPA) outputs ($G_1$); and after the final dense output layer ($G_5$). Our exploration covers gating variants including elementwise and headwise, head-specific and head-shared, as well as additive and multiplicative forms. We find that: **(i)** applying SDPA output head-specific gating ($G_1$) yields the most significant performance improvements (e.g., up to 0.2 PPL reduction and 2 points on MMLU); **(ii)** the SDPA output gating also improves training stability, nearly eliminating loss spikes, enabling larger learning rates and enhancing model scalability.

We identify two factors contributing to the efficacy of gating: **(i) Non-Linearity.** The two consecutive linear layers - the value ($W_v$) and dense ($W_O$) projections - can be rewritten into one low-rank linear projection. Therefore, introducing non-linearity through gating at positions $G_1$ or $G_2$ can increase the expressiveness of this low-rank linear transformation (Sec. 4.1). **(ii) Sparsity.** Although non-linear gating variants consistently enhance performance, we observe that their gains vary. Our analysis further reveals that the pronounced sparsity of the gating scores is another crucial factor, introducing input-dependent sparsity to SDPA outputs (Sec. 4.2). Sparse also gating eliminates the *massive activation* [7] and *attention sink* [8]: the initial tokens have large activation values in the corresponding hidden states (Tab. 4) and disproportionately dominate attention scores (Fig. 2, Sec. 4.3). Previous work [8, 7, 23] explains attention sinks as an accumulation of redundant attention due to non-negative softmax normalization. Empirically, we verify that when *query-dependent sparse gating is applied at the SDPA output*, both our dense and MoE models (trained on 3.5T tokens) exhibit no attention sink. Furthermore, these models demonstrate superior performance in length generalization, achieving a gain of over 10 points on RULER [24](Sec.4.4).

> **Practical Recommendation.** For best results, apply *elementwise SDPA gating $G_1$* (i.e., gating after the attention-weighted value projection) and train with a *moderately increased learning rate*.

## 2 Gated-Attention Layer

### 2.1 Preliminary: Multi-Head Softmax Attention

Given an input $X \in \mathbb{R}^{n \times d_{\text{model}}}$, where $n$ is the sequence length and $d_{\text{model}}$ is the model dimension, the computation of transformer's attention layer [22] could be divided into four stages.

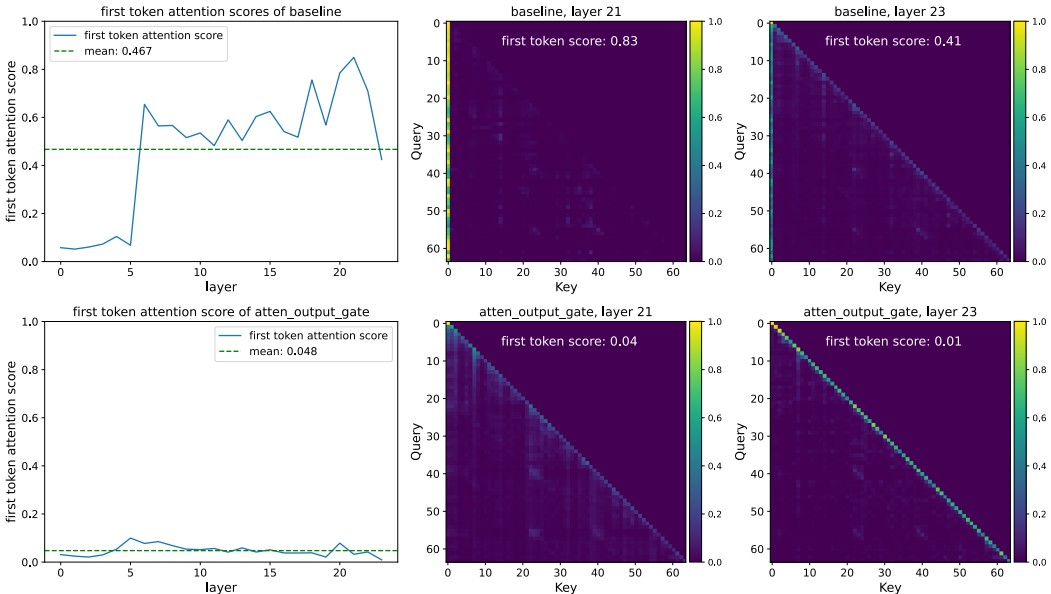

Figure 2: **Left**: Proportion of attention allocated to the initial token per layer (test perplexity dataset). The baseline model suffers from a significant attention sink, with an average of 46.7% of attention scores across layers directed towards the first token. Introducing a gate effectively alleviates this, reducing the proportion to 4.8%. **Right**: Average attention map weights for each head. Layer 21 in the baseline model demonstrates a strong attention sink (83% on the first token), which is substantially reduced by the gate (4%). In the final output layer, the gate amplifies the existing tendency for the model to attend to individual tokens within the sequence.

**QKV Linear Projections:** The input $X$ is linearly transformed into queries $Q$, keys $K$, and values $V$ using learned weight matrices $W_Q, W_K, W_V \in \mathbb{R}^{d_{\text{model}} \times d_k}$ and $Q, K, V \in \mathbb{R}^{n \times d_k}$:

$$Q = XW_Q, \quad K = XW_K, \quad V = XW_V. \tag{1}$$

**Scaled Product Dot-Product Attention (SDPA):** computes attention scores between queries and keys, followed by a softmax normalization. The output is a weighted sum of the values:

$$\text{Attention}(Q, K, V) = \text{softmax}\left(\frac{QK^T}{\sqrt{d_k}}\right)V, \tag{2}$$

where $\frac{QK^T}{\sqrt{d_k}} \in \mathbb{R}^{n \times n}$ represents the scaled dot-product similarity matrix, and softmax$(\cdot)$ ensures the attention weights are no-negative and sum to 1 across each row.

**Multi-Head Concatenation:** In multi-head attention, the above process is repeated for $h$ heads, with each head having its projection matrices $W_q^i, W_k^i, W_v^i$. All heads' outputs are concatenated:

$$\text{MultiHead}(Q, K, V) = \text{Concat}(\text{head}_1, \ldots, \text{head}_h), \tag{3}$$

where $\text{head}_i = \text{Attention}(QW_Q^i, KW_K^i, VW_V^i)$.

**Final Output Layer:** The concatenated SDPA output is passed to the output layer $W_o \in \mathbb{R}^{hd_k \times d_{\text{model}}}$:

$$O = \text{MultiHead}(Q, K, V)W_o. \tag{4}$$

## 2.2 Augmenting Attention Layer with Gating Mechanisms

The gating mechanism is formalized as:

$$Y' = g(Y, X, W_\theta, \sigma) = Y \odot \sigma(XW_\theta), \tag{5}$$

where $Y$ is the input to be modulated, $X$ is another input used to compute the gating scores[1], $W_\theta$ refers to the learnable parameters of gate, $\sigma$ is an activation function (e.g., sigmoid), and $Y'$ is

---

[1]We adopt the hidden states after pre-normalization as $X$.

the gated output. The gating score, $\sigma(XW_\theta)$, effectively acts as a dynamic filter, controlling the information flow from $Y$ by selectively preserving or erasing its features.

This work comprehensively investigates variants of gating mechanisms within the attention layer. Our exploration focuses on five key aspects: **(1) Positions.** We study the effect of applying gating at different positions, as illustrated in Fig. 1(left): (a) after the $Q, K, V$ projections (Equ. 1), corresponding to positions $G_2, G_3, G_4$ in Fig. 1(left); (b) following the SDPA (Equ. 3) outputs ($G_1$). (c) after the final concatenated multi-head attention outputs (Equ. 4, $G_5$). **(2) Granularity.** We consider two levels of granularity for the gating score: (a) Headwise: A single scalar gating score modulates the entire output of an attention head. (b) Elementwise: Gating scores are vectors with the same dimensionality as $Y$, enabling fine-grained, per-dimension modulation. **(3) Head Specific or Shared.** Given the multi-head nature of attention, we further consider: (a) Head-Specific: each attention head has its specific gating scores, enabling independent modulation for each head. (b) Head-Shared: $W_\theta$ and gating scores are shared across heads. **(4) Multiplicative or additive.** For applying gating score to $Y$, we consider (a) Multiplicative Gating: The gated output $Y'$ is computed as: $Y' = Y \cdot \sigma(X\theta)$. (b) Additive Gating: $Y' = Y + \sigma(X\theta)$. **(5) Activation Function.** We mainly consider two common activation functions: SiLU [25] and sigmoid. We only use SiLU for additive gating due to its unbounded output range, and sigmoid only gives scores in $[0, 1]$. Additionally, to further dissect the mechanisms underlying gating's effectiveness, we also consider Identity Mapping or RMSNorm [26] (detailed in Sec 4.1). Unless otherwise specified, we employ head-specific, multiplicative gating utilizing the sigmoid activation function ($\sigma(x) = \frac{1}{1+e^{-x}}$).

## 3 Experiments

### 3.1 Experimental Setups

**Model Architecture and Training Settings** We conduct experiments on both MoE models (15B total parameters with 2.54B activated, 15A2B) and dense models (1.7B total parameters). The 15A2B MoE models utilize 128 total experts with top-8 softmax gating, fine-grained experts [27], global-batch LBL [28], and z-loss [29]. We adopt group query attention (GQA) [30] for the attention part. More detailed model architecture configurations are discussed in Appendix A.2. We train the models on subsets of a 4T high-quality tokens, encompassing multilingual, math, and general knowledge content. A sequence length of 4096 is used. More detailed configurations, such as learning rate and batch size (bsz), will be introduced in each part. Other hyperparameters follow the default values of the AdamW optimizer. Since the parameters and flops introduced by the gating are relatively small, *the wall-time latency introduced by gating is less than 2%*. **Evaluation** We test the few-shots results on popular benchmarks, including, Hellaswag [31] for English, MMLU [32] for general knowledge, GSM8k [33] for math reasoning, HumanEval [34] for coding, C-eval [35] and CMMLU [36] for Chinese proficiency. We also test the perplexity (PPL) on diverse held-out test sets, including domains like English, Chinese, Code, Math, Law and Literature.

### 3.2 Main Results

#### 3.2.1 Gated Attention for MoE models

We first compare different gatings on the training-efficient MoE-15A2B models. All models use a scheduler that warms up to a maximum LR of 2e-3 in 1k steps and decays using cosine to 3e-5. We use a global bsz of 1024, comprising 100k optimization steps. The results are summarized in Tab. 1. To provide a fair comparison, we supplement the vanilla MoE baseline (row 1) with parameter expansion methods, including increasing the number of key-value heads (row 2), increasing the number of query heads (row 3), and increasing both the total and activated number of experts (row 4). These methods introduce a comparable or greater number of parameters than the gating mechanisms.

From Tab. 1, we observe: **(i) SDPA and value output gating are effective**. Inserting gates at the output of SDPA ($G_1$) or the value map ($G_2$) is the most effective, achieving lower PPL and better overall benchmark performance than other variants. We will further investigate why gating at these two positions is effective in Sec 4.2. **(ii) Head-Specific Gating Matters**. Applying headwise gating at $G_1$ and $G_2$ introduces very few additional parameters (less than 2M for the MoE-15A2B model) but still delivers substantial improvements (rows 10 and 11). When sharing gating scores across different attention heads (we average over the query head dimension $q$ to obtain an $n \times d_k$ score from

Table 1: Gating variant performance and results. We train the **15A2B** MoE models on **400B** tokens. $d_k$ is the head dim, $d_{model}$ is the model's hidden dim, and $n$ is the number of tokens. $q$ refers to the number of query heads, $k$ refers to the number of key-value heads. 'Act Func' is the activation function in Eq 5. 'Score Shape' is the gating score shape for an input $X \in \mathbb{R}^{n, d_{model}}$. 'added param' indicates added parameters.

| Method | Act Func | Score Shape | Added Param | Avg PPL | Hellaswag | MMLU | GSM8k | C-eval |
|---|---|---|---|---|---|---|---|---|
| Reference Baselines (Baseline uses $q = 32, k = 4$. All methods use $d_k = 128$.) | | | | | | | | |
| (1) Baseline | - | - | 0 | 6.026 | 73.07 | 58.79 | 52.92 | 60.26 |
| (2) $k = 8$ | - | - | 50 | 5.979 | 73.51 | 59.78 | 52.16 | 62.26 |
| (3) $q = 48$ | - | - | 201 | 5.953 | 73.59 | 58.45 | 53.30 | 59.67 |
| (4) Add 4 Experts | - | - | 400 | 5.964 | 73.19 | 58.84 | 52.54 | **63.19** |
| Gating Position Variants | | | | | | | | |
| (5) SDPA Elementwise $G_1$ | sigmoid | $n \times q \times d_k$ | 201 | **5.761** | 74.64 | **60.82** | **55.27** | 62.20 |
| (6) v Elementwise $G_2$ | sigmoid | $n \times k \times d_k$ | 25 | 5.820 | 74.38 | 59.17 | 53.97 | 61.00 |
| (7) k Elementwise $G_3$ | sigmoid | $n \times k \times d_k$ | 25 | 6.016 | 72.88 | 59.18 | 50.49 | 61.74 |
| (8) q Elementwise $G_4$ | sigmoid | $n \times q \times d_k$ | 201 | 5.981 | 73.01 | 58.74 | 53.97 | 62.14 |
| (9) Dense Output $G_5$ | sigmoid | $n \times d_{model}$ | 100 | 6.017 | 73.32 | 59.41 | 50.87 | 59.43 |
| Gating Granularity Variants | | | | | | | | |
| (10) SDPA Headwise $G_1$ | sigmoid | $n \times q$ | 1.6 | 5.792 | 74.50 | 60.05 | 54.44 | 62.61 |
| (11) v Headwise $G_2$ | sigmoid | $n \times k$ | 0.2 | 5.808 | 74.38 | 59.32 | 53.53 | 62.61 |
| Head-Specific v.s. Head-Shared Gating | | | | | | | | |
| (12) SDPA Head-Shared $G_1$ | sigmoid | $n \times d_k$ | 201 | 5.801 | 74.34 | 60.06 | 53.15 | 61.01 |
| (13) v Head-Shared $G_2$ | sigmoid | $n \times d_k$ | 25 | 5.867 | 74.10 | 59.02 | 53.03 | 60.61 |
| Multiplicative v.s. Additive | | | | | | | | |
| (14) SDPA Additive $G_1$ | SiLU | $n \times q \times d_k$ | 201 | 5.821 | **74.81** | 60.06 | 53.30 | 60.98 |
| Activation Variants | | | | | | | | |
| (15) SDPA Elementwise $G_1$ | SiLU | $n \times q \times d_k$ | 201 | 5.822 | 74.22 | 60.49 | 54.59 | 62.34 |

the original $n \times q \times d_k$), the benchmark improvements are smaller than those achieved by headwise gating (row 12 v.s. 10, 13 v.s. 11). This underscores the importance of applying distinct gating scores for different attention heads. **(iii) Multiplicative Gating is Preferred**. Additive SDPA output gating underperforms the multiplicative one, although it shows improvements over the baselines. **(iv) Sigmoid Activation is Better**. Replacing the activation function in the most effective gating configuration (row 5) with SiLU (row 15) leads to less improvement.

Overall, adding gating at the value layer ($G_2$) and SDPA output ($G_1$) reduces PPL by more than 0.2, outperforming various parameter-expanding baselines. However, gating at $G_1$ achieves better PPL and benchmark results. As long as different heads receive distinct gating scores, the granularity of gating and the choice of activation function have relatively minor impacts. We will further analyze the reasons behind these observations in Analysis (Sec 4.2).

### 3.2.2 Gated Attention for Dense Models.

We also conduct experiments on dense models following [37] to validate SDPA output sigmoid gating. When using gating, we reduce the width of FFN to maintain the parameter size. Most experiments use optimized hyperparameters for the baseline. For instance, for the 1.7B model trained on 400B tokens, we use a maximum LR of 4e-3 and a bsz of 1024. For training on 3.5T tokens, we increase the maximum LR to 4.5e-3 and the bsz to 2048. Prior work has established that while increased network depth, large learning rates, and large batch sizes can significantly improve model performance [38, 39, 40] and distributed training efficiency, they often introduce training instabilities [39, 41, 42]. We observe that applying gating largely reduces the loss spikes [43, 42] during training (Fig. 2 right), suggesting a promising role for gating in enhancing training stability. Therefore, we introduce another setting characterized by an increased number of layers, a higher maximum learning rate, and a larger batch size to further probe gating's stabilizing effects.

Tab. 2 reveals that: **(i) Gating is effective across various settings** Across various model configurations (row 1 v.s. 2, 5 v.s. 8), training data (row 3 v.s. 4), and hyperparameters (row 11 v.s. 13), SDPA output gating consistently yields benefits. **(ii) Gating improves stability and facilitates scaling.** Under the 3.5T token setting, gating improves training stability, largely reducing the loss spike (Fig. 1, right). When increasing the maximum LR, baselines encounter convergence issues (row 6, 12). While adding sandwich norm [44] restores convergence, the improvement is negligible. In contrast, increasing the maximum LR in models with gating results in a noticeable improvement.

Table 2: Performance of different methods with varying learning rates, batch sizes and model configurations. 'SDPA' refers to the sigmoid gating after SDPA in Eq 3, and 'sandwitch norm' [44] indicates normalizing attention/ffn outputs before adding them to the residual. When using gating, we reduce FFN's width so that all methods have the same number of parameters. '-' means the model diverges during training.

| Method | Max LR | Avg PPL | HumanEval | MMLU | GSM8k | Hellaswag | C-eval | CMMLU |
|---|---|---|---|---|---|---|---|---|
| 28 Layer, 1.7B Parameters, **400B Tokens**, Batch Size=1024 | | | | | | | | |
| (1) Baseline | $4.0 \times 10^{-3}$ | 7.499 | 28.66 | 50.21 | 27.82 | 64.94 | 49.15 | 49.52 |
| (2) SDPA Elementwise | $4.0 \times 10^{-3}$ | **7.404** | **29.27** | **51.15** | **28.28** | **65.48** | **50.72** | **50.72** |
| 28 Layer, 1.7B Parameters, **3.5T Tokens**, Batch Size=2048 | | | | | | | | |
| (3) Baseline | $4.5 \times 10^{-3}$ | 6.180 | 34.15 | 59.10 | 69.07 | 68.02 | 68.19 | 64.95 |
| (4) SDPA Elementwise | $4.5 \times 10^{-3}$ | **6.130** | **37.80** | **59.61** | **70.20** | **68.84** | **68.52** | **65.76** |
| 48 Layer, 1.7B Parameters, **400B Tokens**, Batch Size=1024 | | | | | | | | |
| (5) Baseline | $4.0 \times 10^{-3}$ | 7.421 | 28.05 | 52.04 | 32.98 | 65.96 | 51.11 | 51.86 |
| (6) Baseline | $8.0 \times 10^{-3}$ | 9.195 | 21.34 | 44.28 | 15.24 | 57.00 | 43.11 | 42.63 |
| (7) Baseline+Sandwich Norm | $8.0 \times 10^{-3}$ | 7.407 | 30.49 | 52.07 | 32.90 | 66.00 | 52.04 | 51.72 |
| (8) SDPA Elementwise | $4.0 \times 10^{-3}$ | **7.288** | **31.71** | 52.44 | 32.37 | 66.28 | 52.06 | 52.29 |
| (9) SDPA Headwise | $4.0 \times 10^{-3}$ | 7.370 | 31.10 | 53.83 | 34.12 | 65.59 | **55.07** | 52.38 |
| (10) SDPA Elementwise | $8.0 \times 10^{-3}$ | 7.325 | 31.10 | **54.47** | **36.62** | **66.40** | 53.91 | **53.80** |
| 48 Layer, 1.7B Parameters, **1T Tokens**, Batch Size=4096 | | | | | | | | |
| (11) Baseline | $5.3 \times 10^{-3}$ | 7.363 | 29.88 | 54.44 | 32.22 | 65.43 | 53.72 | 53.37 |
| (12) Baseline | $8.0 \times 10^{-3}$ | - | - | - | - | - | - | - |
| (13) SDPA Elementwise | $5.3 \times 10^{-3}$ | 7.101 | **34.15** | 55.70 | 36.69 | 67.17 | 54.51 | 54.68 |
| (14) SDPA Elementwise | $8.0 \times 10^{-3}$ | **7.078** | 31.71 | **56.47** | **39.73** | **67.38** | **55.52** | **55.77** |

In summary, we identify SDPA element-wise gating as the most effective method to augment the attention mechanism. Incorporating the SDPA output gate enables stable training under larger learning rates and batch sizes—regimes where the baseline often becomes unstable. This suggests that the optimal hyperparameter configuration shifts when using gating. *In practice, one effective way to leverage the gate is to start from the baseline's optimal batch size and moderately increase the learning rate.* Further jointly tuning batch size and learning rate may yield additional gains.

# 4 Analysis: Non-Linearity, Sparsity, and Attention-Sink-Free

In this section, we conduct a series of experiments to explore why such a simple gating mechanism can yield significant improvements in performance and training stability. Here are the takeaways according to our analysis: (1) Gatings enhancing non-linearity consistently lead to performance gains (Sec 4.1); (2) The most effective SDPA elementwise gate introduces strong input-dependent sparsity (Sec 4.2), which then helps to eliminate the 'massive activation' and 'attention sink' phenomenon.

## 4.1 Non-linearity Improves the Expressiveness of Low-Rank Mapping in Attention

Inspired by prior works that utilize group norm for the SDPA output [14, 45], with the same setting in Sec. 3.2.1, we apply RMSNorm [26] independently to the

Table 3: Performance of different (non)-linearity augmentations.

| Method | Activation Function | Avg PPL | Hellaswag | MMLU | GSM8k | C-eval |
|---|---|---|---|---|---|---|
| (1) Baseline | - | 6.026 | 73.07 | 58.79 | 52.92 | 60.26 |
| (2) SDPA Elementwise Gate | Sigmoid | **5.761** | 74.64 | **60.82** | **55.27** | **62.20** |
| (3) v Elementwise Gate | Sigmoid | 5.820 | 74.38 | 59.17 | 53.97 | 61.00 |
| (4) SDPA Additive Gate | SiLU | 5.821 | **74.81** | 60.06 | 53.30 | 60.98 |
| (5) SDPA GroupNorm | RMSNorm | 5.847 | 74.10 | 60.15 | 53.75 | 61.14 |
| (6) SDPA SiLU | SiLU | 5.975 | 73.34 | 59.55 | 53.19 | 60.90 |
| (7) SDPA Additive Gate | Identity | 5.882 | 74.17 | 59.20 | 52.77 | 59.86 |

output of each attention head before concatenation. As shown in Tab. 3 row 5, applying RMSNorm, which introduces almost no additional parameters, also leads to a significant reduction in PPL.

In multi-head attention, the output of the $i$-th token, corresponding to the $k$-th head, can be expressed:

$$o_i^k = (\sum_{j=0}^{i} S_{ij}^k \cdot X_j W_V^k) W_O^k = \sum_{j=0}^{i} S_{ij}^k \cdot X_j (W_V^k W_O^k), \quad (6)$$

where $W_O^k$ is the parameters of the output layer $W_O$ corresponding to the $k$-th head[2]. Here, $S_{ij}^k$ is the attention score of the $i$-th token attending to the $j$-th token in the $k$-th head, $X_j$ is the input to the attention for token $j$, and $X_j W_V^k$ is the value output of token $j$ in the $k$-th head. From Equ. 6, *we can merge $W_V^k W_O^k$ into one low-rank linear mapping applied over all $X_j$ as $d_k < d_{model}$*. With GQA, $W_V$ is shared among heads within the same group, further diminishing the expressiveness.

Given that adding non-linearity between two linear mappings can improve their expensiveness [46], we have two modifications to mitigate the low-rank problem:

$$o_i^k = \left( \sum_{j=0}^{i} S_{ij}^k \cdot \text{Non-Linearity-Map}(X_j W_V^k) \right) W_O^k, \tag{7}$$

$$o_i^k = \text{Non-Linearity-Map} \left( \sum_{j=0}^{i} S_{ij}^k \cdot X_j W_V^k \right) W_O^k. \tag{8}$$

Notably, adding gating at the $G_2$ (Tab. 3 row 3) position corresponds to the first modification (Equ. 7), while adding gating (row 4) or group normalization (row 5) at the $G_1$ position corresponds to the second (Equ. 8). This also explains why adding gating or normalization at the $G_5$ position after $W_O$ has no effect (Tab. 1 row 9) — it does not address the lack of non-linearity between $W_V$ and $W_O$.

For additive gating at $G_1$, the output of gating passes through SiLU (Tab. 3 row 4), also introducing some non-linearity, which explains the observed performance gains, albeit smaller than those achieved by multiplicative gating. Based on these insights, we conduct two additional experiments: **(i)** Adding SiLU only at the $G_1$ position without introducing additional parameters (Tab. 3 row 6). Notice this simple modification also leads to a modest reduction in PPL, but most benchmark scores remain unchanged. **(ii)** Removing SiLU from additive gating, such that the output of $X_j$ after gating is directly added at the $G_1$ position (Tab. 3 row 7). This further diminishes the gains of addictive gating.

In summary, the enhanced performance associated with effective gating variants is likely attributable to the introduction of non-linearity between $W_V$ and $W_O$. Although applying gating at positions $G_1$ and $G_2$ can can both introduce this non-linearity, these applications yield differing performance gains. This observed difference motivates us to further analyze the impacts of gating at these two positions.

## 4.2 Gating Introduces Input-Dependent Sparsity

We analyze the gating scores (Tab. 1, 'Gate Score' column) of models with gating applied at the value ($G_2$) and SDPA output ($G_1$) positions, evaluated on the test language modeling data. The mean gating scores for all layers are presented in Table 4, with the score distributions visualized in Fig. 3 (layer-wise scores in Appendix A.3). Key observations include:

**(i) Effective Gating Scores are Sparse.** SDPA output gatings (Elementwise/headwise) exhibit the lowest mean gating scores. Furthermore, the SDPA output gating score distribution shows a high concentration near 0, indicating substantial sparsity, consistent with its superior performance.
**(ii) Head-Specific Sparsity Matters.** Enforcing shared gating scores across attention heads increases the overall gating scores and diminishes performance gains. Observations (i) and (ii) underscore the importance of *head-specific gating*, aligning with previous research demonstrating that individual attention heads capture distinct aspects of the input [47, 48, 49, 50].

**(iii) Query-Dependency Matters.** The scores for value gating ($G_2$) are higher than those for SDPA output gating ($G_1$), and the performance is inferior. This suggests that gating score sparsity is more effective when query-dependent rather than determined by the key and value. Specifically, SDPA output gating scores are derived from the hidden states corresponding to the current query (e.g. the Non-Linearity-Map in Eq 8 depends on $X_i$), whereas value gating scores are derived from hidden states associated with past keys and values (e.g. the Non-Linearity-Map in Eq 7 depends on each $X_j$). This implies that *gating score sparsity may filter out irrelevant contextual information for the query*. To further validate the importance of query-dependency, we introduce input-independent gating by zero-initializing learnable parameters ($q \times d_k$), applying a sigmoid function, and multiplying it with the SDPA output. As shown in row (6), input-independent gating improves upon the baseline, likely due to the introduction of non-linearity. Moreover, the high gating scores reinforce that effective sparsity should be input-dependent.

---

[2]Note that concatenating outputs from different heads and then multiplying with $W_O$ is equivalent to multiplying each head's output with its corresponding $W_O^k$ before concatenation

Table 4: Performance of different gating methods with varying activation functions and average gate scores. 'Act-Func' refers to the activation function used for computing the gating scores, while 'M-Act' denotes the rounded maximum activation values of the hidden states output by each layer of the model. Additionally, 'F-Attn' represents the attention score of the first token, with higher values indicating more pronounced 'attention sink'.

| Method | Act-Func | Gate Score | M-Act | F-Attn | PPL | Hellaswag | MMLU | GSM8k |
|---|---|---|---|---|---|---|---|---|
| (1) Baseline | - | - | 1053 | 0.467 | 6.026 | 73.07 | 58.79 | 52.92 |
| (2) SDPA Elementwise Gate | Sigmoid | 0.116 | 94 | 0.048 | **5.761** | **74.64** | **60.82** | **55.27** |
| (3) SDPA Headwise Gate | Sigmoid | 0.172 | 98 | 0.073 | 5.792 | 74.50 | 60.05 | 54.44 |
| (4) SDPA Elementwise Head-shared Gate | Sigmoid | 0.271 | 286 | 0.301 | 5.801 | 74.34 | 60.06 | 53.15 |
| (5) v Elementwise Gate | Sigmoid | 0.221 | 125 | 0.297 | 5.820 | 74.38 | 59.17 | 51.33 |
| (6) SDPA Input Independent Gate | Sigmoid | 0.335 | 471 | 0.364 | 5.917 | 73.64 | 59.02 | 52.40 |
| (7) SDPA Elementwise Gate | NS-sigmoid | 0.653 | 892 | 0.451 | 5.900 | 74.05 | 60.05 | 52.75 |

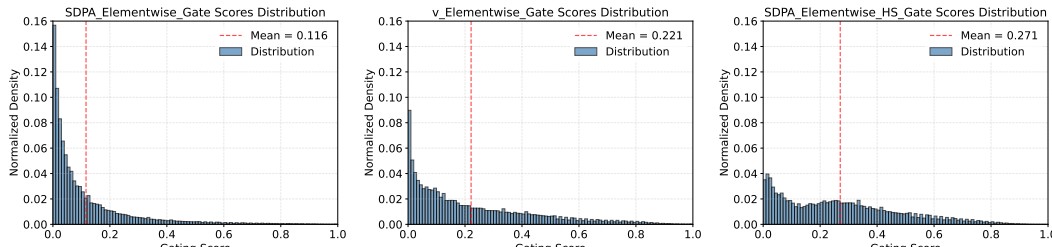

Figure 3: Gating score means and distributions for SDPA elementwise (Left), value Elementwise (Middle), and SDPA elementwise with head-shared gating (Right). Most gating scores are less than 0.5, indicating that the gating scores are sparse. Among them, the SDPA output gating score exhibits the strongest sparsity.

**(iv) Less Sparse Gating is Worse.** To further validate the importance of gating score sparsity, we reduce sparsity from the gating formulation. Specifically, we replace the sigmoid function with a modified Non-Sparse (NS) version:

$$\text{NS-sigmoid}(x) = 0.5 + 0.5 \cdot \text{sigmoid}(x), \tag{9}$$

which constrains the gating scores between [0.5, 1.0]. This ensures introducing non-linearity while removing gating score sparsity. As shown in Tab. 4 row (7), the gains of NS-sigmoid gating are inferior to those of SDPA output sigmoid gating. In Appendix A.3, we provide a more detailed discussion on how sparse gating scores affect the sparsity (the proportion of values below the threshold) in SDPA hidden states. We will discuss the impact of different sparsity levels on model behavior, including reducing the 'attention sink', in the next section.

### 4.3 SDPA Output Gating Reduces Massive Activation and Attention-Sink

Based on the observation that gating introduces sparsity to the SDPA output in an input-dependent manner, we hypothesized that this mechanism can filter out context irrelevant to the current query token, thereby mitigating the attention sink [8, 7]. Correspondingly, early work [51] finds that in explicit top-$k$ sparse attention—where only the most relevant tokens are attended to—the attention sink phenomenon does not occur. To verify this, we analyze the distribution of attention scores (averaged over all heads) and the proportion of attention scores allocated to the first token (Fig. 2, Tab. 4, 'F-Attn' column). Massive activations [7] (large values in hidden states) are belied to lead to the attention sink to their corresponding tokens. Inspired by this, we also compute the mean of the maximum hidden state activations across layers, as shown in the 'M-Act' column of Tab. 4. More detailed layer-wise results are provided in the Appendix A.4.

We can observe: **(i)** Head-wise and element-wise query-dependent sigmoid gating at the SDPA output ($G_1$) largely reduces the attention score allocated to the first token and decreases massive activations. **(ii)** Enforcing shared gating scores across heads or applying gating only after the value projection ($G_2$) decreases massive activations, but does not reduce attention scores to the first token. This reinforces the importance of head-specific gating and suggests that *massive activations are not a necessary condition for attention sinks*. **(iii)** Reducing the input-dependence of gating (row 6) or using NS-sigmoid to reduce sparsity (row 7) intensifies both massive activations and attention sink.

Collectively, these observations indicate that *input-dependent, head-specific gating of the SDPA output introduces significant sparsity, thereby mitigating the attention sink*. Furthermore, sparsity in

the SDPA outputs reduces massive activations within the model, with increased sparsity leading to smaller activations. *This may explain the improved training stability with gating: by reducing massive activations, the model is less susceptible to numerical errors during BF16 training [52].* We also observe that massive activations originate primarily from early layers (e.g., layer 5), where the FFN outputs large values, consistent with [53]. Once added to the residual stream, these activations are propagated through subsequent layers via the pre-norm mechanism. This aligns with the effectiveness of sandwich normalization [44] in enhancing training stability (Table 2, row 7): applying LayerNorm to the FFN output prevents these large activations from entering the residual stream.

## 4.4 SDPA Output Gating Facilitates Context Length Extension

Based on the attention-sink-free pattern, we evaluate the SDPA gating's effect in the long-context setting. Specifically, we extend the context length for the models trained on 3.5T tokens. We increase the RoPE [54] base

Table 5: Performance of different methods across varying sequence lengths. 'YaRN Extended' indicates the expanded context length variant. '(values)' indicate the performance declines after extending the context length.

| Method | 4k | 8k | 16k | 32k | 64k | 128k |
|---|---|---|---|---|---|---|
| Baseline | 88.89 | 85.88 | 83.15 | 79.50 | - | - |
| SDPA-Gate | 90.56 | 87.11 | 84.61 | 79.77 | - | - |
| YaRN Extended | | | | | | |
| Baseline | 82.90(-6.0) | 71.52(-14.4) | 61.23(-21.9) | 37.94(-41.56) | 37.51 | 31.65 |
| SDPA-Gate | 88.13(-2.4) | 80.01(-7.1) | 76.74(-7.87) | 72.88(-6.89) | 66.60 | 58.82 |

from 10k to 1M and continue training on data with a sequence length of 32k for an additional 80B tokens. This gives us models with a context length of 32k. Subsequently, we use YaRN [55] to extend the context length to 128k. We evaluate models on the RULER benchmark [24] and summarize results in Tab. 5. We observe the following: **(i)** Under the 32k setting, models with gating slightly outperform the baseline. This suggests that within the training length, the attention sink phenomenon may not hurt the model's long-context performance. **(ii)** When the context length is extended to 128k using YaRN, both the baseline and gated models experience a decline within the original 32k range. This observation is consistent with previous works on extending context length by modifying RoPE [56, 55, 57]. Even though the decline is less pronounced for models with gating. **(iii)** At context lengths of 64k and 128k, the gated attention models outperform the baseline signifantly. From these observations, we hypothesize that adding gating helps the model adapt to the context-length extension. A possible explanation is that baseline models rely on attention sinks to adjust the distribution of attention scores. [57] derives the effects of changing the RoPE based on the attention and hidden state distributions. When techniques like YaRN are applied to modify the RoPE base, the attention sink pattern may struggle to adapt in a training-free manner, leading to a noticeable drop in performance. In contrast, models with gating primarily rely on input-dependent gating scores to control information flow, making them more robust to such changes.

# 5 Related Works

## 5.1 Gating in Neural Networks

Gating mechanisms have been widely adopted in neural networks. Early works such as LSTMs [1] and GRUs [9] introduce gates to regulate information flow across time steps, addressing gradient vanishing/exploding issues by selectively retaining or discarding information. Highway Networks [2] extend this concept to feedforward networks, enabling the successful training of very deep architectures. SwiGLU [25] introduce gating mechanisms into transformer FFN layers, enhancing their expressive power and becoming a standard component in many open-source LLMs [58, 37].

Several works on state-space models [3, 10, 59] and Linear Attention, such as FLASH [4], RetNet [14], Lightning Attention [17, 60, 61], and Gated Delta Networks [18], also incorporate gating modules to controlinformation of token-mixer modules. AlphaFold2 [5] and Forgetting Transformer [6] introduce gating mechanisms to the output of softmax attention. GaAN [11] uses gating to control each attention head's importance for learning on graphs. Some works [62, 13, 63, 15, 16, 64, 65, 66] also apply operations similar to gating to augment softmax attention. Attention on Attention (AoA) [12] also modulates the attention output with a sigmoid gating, depending on the query. Although these works demonstrate the effectiveness of gating, a comprehensive understanding of its precise mechanisms and the reasons behind its effectiveness still needs exploration. This could contribute to

a broader appreciation of gating's importance beyond RNNs and facilitate designs that better leverage gating's unique advantages. For example, while Switch Heads [20, 19], NSA [21], and MoSA [67] employ sigmoid-based gating [68] for selection, further investigation into isolating gating's specific contribution could offer valuable insights. Comparisons with baselines incorporating similar gating mechanisms in standard transformers could offer a more refined perspective on the effectiveness of their proposed selection mechanisms. The work most closely related to ours is Quantizable Transformers [69], which also finds that applying gating in softmax attention alleviates extreme attention concentration and outliers in hidden states in encoder models like BERT and ViT. While this work primarily leverages gating to eliminate outliers for model quantization, we provide a detailed analysis of various gating variants, uncovering their benefits through enhanced non-linearity and sparsity, as well as improved training stability. Building on these insights, we scale up gated attention models, demonstrating gating's broad applicability and impact.

## 5.2 Attention Sink

StreamingLLM [8] formally identifies 'attention sink', in which specific tokens receive large attention scores. Similarly, in ViT, some redundant tokens act as 'registers' to store attention scores [70]. Later, Massive Activation [7] shows that excessive attention scores are assigned to tokens associated with massive activation values. However, our work reveals that value output ($G_2$) gating eliminates massive activations, yet attention sinks persist, indicating that massive activations are necessary for attention sinks. Similarly, attention sinks are characterized as non-informative 'key biases' that store redundant attention scores, arguing that softmax's inherent normalization dependency drives this behavior [23]. Experimental attempts to modify softmax attention, such as replacing softmax with unnormalized sigmoid attention [71, 23], using explicit top-k sparse attention [51], adding softmax attention gate or clip [69], calibrating attention scores [72], and modifying softmax computation [73] and denominator [74], show promise in mitigating attention sinks. Another stream of works try to move the sink tokens from input tokens to manually added components, like 'registers' [70], 'meta tokens' [75] and learnable 'sink' [76]. Our work shows that sparse gating after SDPA eliminates attention sinks in both dense (1B-parameter) and MoE (15B-parameter) models, even when trained on 3.5T tokens. Furthermore, we uncover the potential of eliminating attention sinks to benefit context-length extension.

## 6 Conclusion and Limitations

This work systematically investigates gating mechanisms in softmax-attention, revealing their significant impact on performance, training stability, and attention dynamics. This simple mechanism enhances non-linearity, introduces input-dependent sparsity, and eliminates 'attention sink'. Additionally, gating facilitates context length extension, allowing models to generalize effectively to longer sequences without retraining. We will release the 'attention-sink-free' models, providing a foundation for future research into attention mechanisms.

The broader implications of non-linearity on the dynamics of attention and the overall training process remain under-explored. We don't provide a theoretical explanation for how attention sinks influence the model's ability to generalize to longer sequences.

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

# A  Supplement Material

## A.1  Switch Head Baselines

In this section, we present detailed experiments related to Switch Heads. The Switch Head paper demonstrates that introducing sparse activation in attention—where each token selects the top-k experts from a pool of key/value/output experts via learnable sigmoid routing—enables the model to achieve comparable results to the baseline. This suggests that, within the Switch Head framework, both expert parameters and activated parameters are beneficial, with more being better under the same total parameter budget.

Table 6: Performance of different switch head methods with varying parameter additions and configurations. 'switch kv' and 'switch v' refer to introducing selective computing in key-value and value components, respectively. 'Switch kv, 8top8' means there are 8 key and value map experts, and each token select top8 experts. Notice 'Switch v, 1top1' is equivalent to v Headwise Gate in Tab. 1 row (11).

| Method | Added Param (M) | PPL | MMLU | GSM8k | Hellaswag | C-eval |
|---|---|---|---|---|---|---|
| (1) Baseline (q32, kv4) | - | 6.026 | 58.79 | 52.92 | 73.07 | 60.26 |
| (2) Switch kv, 8top8 | 38 | 5.847 | 59.17 | 52.54 | 73.32 | 61.01 |
| (3) Switch kv, 4top4 | 13 | 5.935 | 58.14 | 53.27 | 73.75 | 59.67 |
| (4) Switch v, 4top4 | 13 | 5.820 | 59.02 | 52.77 | 73.34 | 61.74 |
| (5) Switch v, 8top2 | 25 | 5.870 | 59.10 | 53.53 | 74.17 | 62.34 |
| (6) Switch v, 1top1 | 3 | 5.808 | 59.32 | 53.53 | 74.38 | 62.61 |

Looking at the results in Tab. 6, we observe an interesting trend: while increasing the number of activated kv experts (with the same expert parameter settings) appears to offer some improvement in PPL (row 4 vs. 5), the gains in overall benchmark performance are less pronounced. Notably, the best results for both benchmark scores and PPL were achieved by 'Switch v 1top1' (row 6), which, as mentioned earlier, is analogous to applying sigmoid gating directly to the output of the value layer. These findings raise an intriguing question about the primary driver of the performance improvements observed in these experiments. It suggests that the introduction of gating itself plays a significant role in the effectiveness of this approach.

## A.2  More Discussion on Sparse Gating Score

In our experiments, we evaluate three distinct LLM architectures: two dense variants with 1.7 billion parameters and one sparse MoE model with an effective size of approximately 15 billion parameters (denoted as 15A2B). The two dense models differ primarily in depth and hidden dimension: the *1.7B-28 layer* model uses 28 transformer layers with a hidden size of 2048, while the *1.7B-48 layer* variant employs 48 layers but a reduced hidden size of 1536 to maintain a comparable parameter count. Both dense models tie input and output embeddings and apply query-key normalization for training stability. Full architectural hyperparameters are summarized in Table 7. To support reproducibility and future research, we plan to open-source both 1.7B dense models.

## A.3  More Discussion on Sparse Gating Score

In this section, we analyze the impact of gating score sparsity on attention output. First, we examine the mean values of SDPA output before and after applying gating to the hidden states. Specifically, we calculated the mean absolute values of $Y$ and $Y'$ before and after $G_1$ at each layer, as shown in Fig. 4. We also included results from a baseline without gating for comparison. The results indicate that: (1) after gating, the mean value of hidden states decreased from 0.71 to 0.05, corresponding to the generally small gating scores; (2) the gated hidden states closely resemble the baseline, suggesting that gating might serve a similar function as attention sink in filtering out irrelevant information.

We further analyze the proportion of hidden states below certain thresholds before and after gating, as shown in Fig 5. The results reveal that: (1) after gating, the sparsity in hidden states significantly increases across different thresholds. Since the mean gating scores are already small, multiplying hidden states by a small number naturally pushes some values below the threshold. Therefore, (2) we further multiply the pre-gating hidden states by the average gating score and observed that the increase in sparsity is smaller than with original gating. This suggests that sparse gating scores enhance sparsity in hidden states.

Table 7: Architectural specifications of the target LLMs used in our experiments. The 15A2B model is a Mixture-of-Experts (MoE) architecture with 128 experts and top-8 routing. All models use a head dimension of 128 and apply query-key normalization. Embedding weights are tied in the dense models but not in the MoE model.

| Model | 1.7B-28 layer | 1.7B-48 layer | 15A2B MoE |
|---|---|---|---|
| Layers | 28 | 48 | 24 |
| Query Heads | 16 | 16 | 32 |
| Key / Value Heads | 8 | 8 | 4 |
| Head Dimension | 128 | 128 | 128 |
| Tie Embedding | Yes | Yes | No |
| QK Normalization | Yes | Yes | Yes |
| Hidden Size | 2048 | 1536 | 2048 |
| FFN Size | 6144 | 4608 | 768 |
| Number of Experts | – | – | 128 |
| Top-$k$ | – | – | 8 |

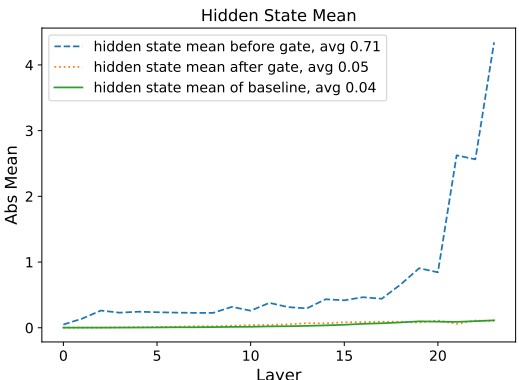

Figure 4: Mean absolute values before and after gating. The baseline and post-gating values are similar.

### A.4   Layerwise Massive Activations and Attention Sinks

In this section, we compare and analyze the presence of massive activations and attention sinks (the attention score of the first token) within the model. From the results, we observe the following:

For the baseline (row 1), the output of the 6th layer's FFN contains massive activations, which are subsequently added to the residual stream, causing large activations to persist in the residuals of subsequent layers. Correspondingly, significant attention sink phenomena emerge starting from the 6th layer. After applying gating to the SDPA output (row 2), the outputs of the earlier layers in the network remain relatively small overall, with massive activations growing gradually as the layer depth increases. Notably, no significant attention sink phenomenon is observed in any layer of the network.

When gating is applied only at the value layer (row 3), the model exhibits massive activations similar to row 2. However, a certain degree of attention sink phenomenon persists. This indicates that massive activations are not a necessary condition for the emergence of attention sinks. When enforcing shared gating scores across different heads (row 4) or modifying the activation function of gating to suppress sparsity (row 5), the sparsity introduced by gating is reduced. In these cases, both massive activations and attention sinks become comparable to those observed in the baseline.

These observations suggest that introducing sufficient sparsity within the attention mechanism may help mitigate the occurrence of massive activations. However, further investigation is needed to fully understand the interplay between sparsity, massive activations, and attention sinks, particularly in the context of scaling to deeper and larger models.

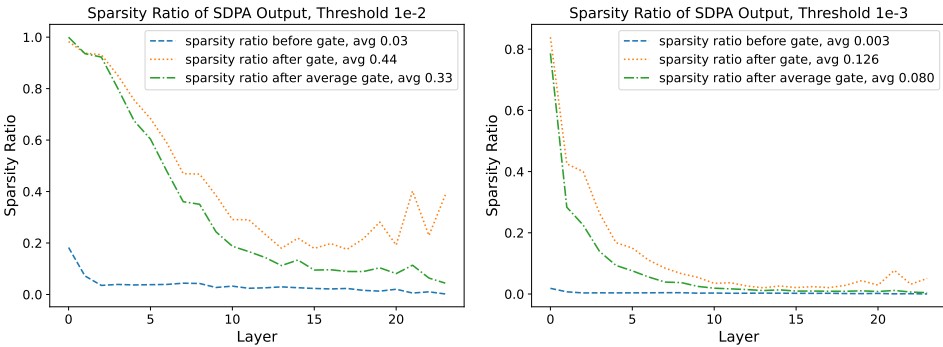

Figure 5: Proportion of SDPA output values below threshold after gating (Left: 1e-2, Right: 1e-3). We also include sparsity measurements obtained by multiplying the average gating score with pre-gating hidden states.

## A.5  More Layerwise Gating Scores

In this section, we analyze the distribution of gating scores under two additional constraints while using SDPA output gating as the baseline (row 1, elementwise/headwise): (1) enforcing the same gating score across different heads (row 2, left), and (2) restricting the minimum value of the gating scores (row 2, right). When enforcing shared gating scores across different heads, the gating scores for most layers increase. This indicates that different heads require different sparsity, highlighting the importance of head-specific gating mechanisms.

## A.6  Other Attempt to Stabilize Training

We observe that both the addition of sandwich normalization [44] and gating mechanisms eliminate massive activations while improving training stability. This prompts us to explore whether simpler methods could prevent large activations within residuals. Specifically, we introduce a clipping operation to constrain the outputs of attention and FFN layers before they enter the residual connection, limiting their values to the range (-clip, clip). However, we find that regardless of whether the clip value was set to 300 or 100, the model still encounters convergence issues at a learning rate of 8e-3. This suggests that the instability in pre-norm model training is not solely due to large activations within residuals. It is likely that any layer producing large outputs can lead to stability problems, indicating the need for further investigation into the root causes of training instability.

## A.7  Adding Gating in Continue Training

We also experiment with incorporating an attention output gating mechanism during the continued training phase. However, we find that this approach neither mitigates the massive activations and attention sinks already present in the model nor significantly affects the final performance. We believe this is largely because the effectiveness of gating mechanisms stems primarily from their affects on the model's training dynamics, and thus their impact is limited when applied to models that are not trained from scratch.

## A.8  Broader Impacts

This work focuses on improving the efficiency, stability, and context-handling capabilities of LLMs. The potential positive societal impacts include: Improved Accessibility: More efficient and stable training methods can lower the computational cost of developing and deploying LLMs, potentially making them more accessible to researchers and organizations with limited resources. Enhanced Performance in Long-Context Applications: The ability to handle longer contexts more effectively can lead to improvements in applications such as document summarization, question answering, and code generation, where understanding relationships across extended text is crucial. Advancements in Model Scalability: Our findings offer insights and practical guidance for designing advanced models, contributing to ongoing efforts to scale LLMs while maintaining or improving their performance and stability.

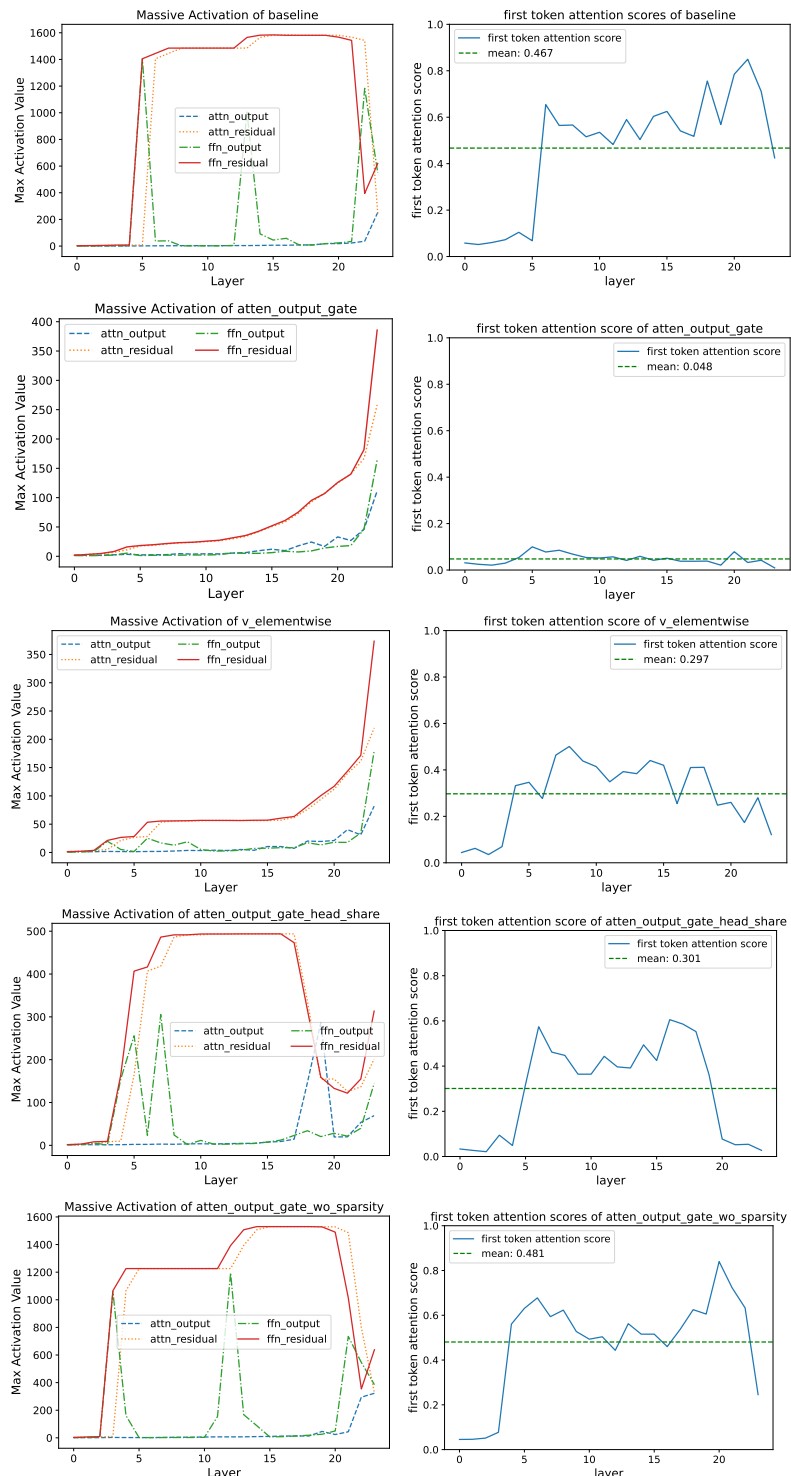

Figure 6: Comparison of massive activations and attention sink phenomena across different gating configurations. Row 1 (Baseline): Significant massive activations and attention sinks emerge after the 6th layer. Row 2 (SDPA Gating): Reduced activations and no attention sinks observed. Row 3 (Value Layer Gating): Similar activations to Row 2 but with residual attention sinks. Rows 4–5 (Reduced Sparsity via cross-head share and NS-sigmoid): Massive activations and attention sinks resemble the baseline.

## A.9 Licenses for Existing Assets

We conduct experiments based on Megatron-LM. We acknowledge the original authors and contributors of Megatron-LM and respect their licensing terms. The specific license terms are available at

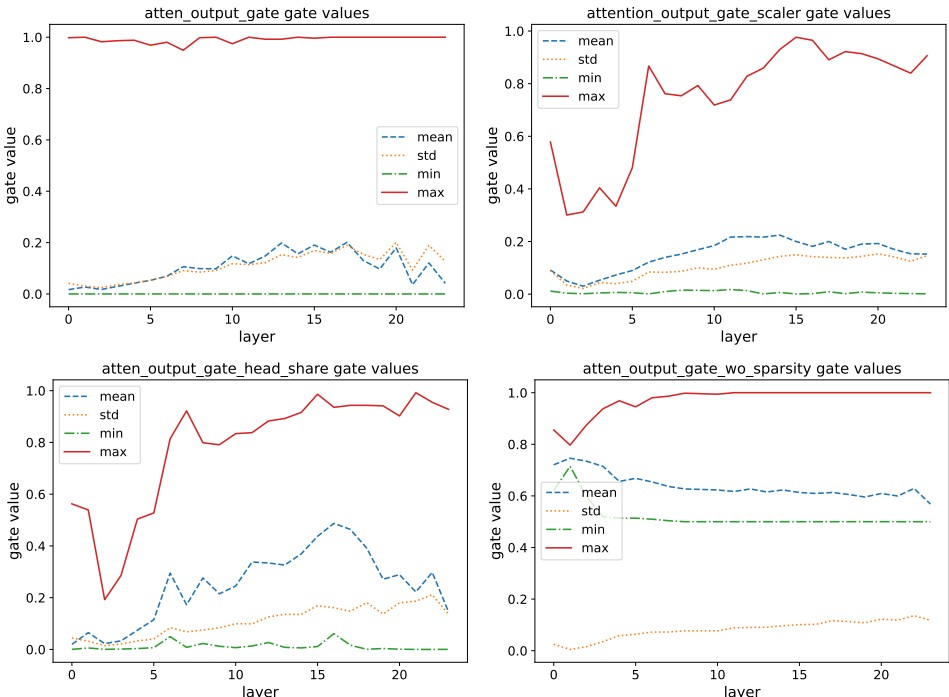

Figure 7: Distribution of gating scores under different constraints for SDPA output gating variants.

https://github.com/NVIDIA/Megatron-LM/blob/main/LICENSE. Any released code implementing our proposed gated attention mechanism will be released under the MIT License, allowing for open research and further development.

