# OpenReview forum: "Gated Attention for Large Language Models: Non-linearity, Sparsity, and Attention-Sink-Free"
_NeurIPS.cc/2025/Conference — NeurIPS 2025 oral_

### Official Review · Reviewer_pXLA · 2025-06-28

**Clarity:** 3
**Significance:** 4
**Originality:** 3
**Rating:** 5
**Confidence:** 3

**Summary:**

This work aims to propose an improvement on the self-attention module widely used in LLMs, introducing gating mechanisms to the typical self-attention layer. The work systematically investigates the  impact of gating mechanisms from a range of perspectives, including placing gates across various positions, different granularity, headspecific or shared, etc., which provides a comprehensive comparison between gating mechanisms with these nuances.
Experimental results  across popular benchmarks indicate that this simple modification can improve model performance and training stability. Particularly, SDPA Output gating can reduce massive activation and attention-Sink, creating more balanced roles for weights and attention scores. Additionally, this gating helps improve the performance on tasks involving context length extension.

**Questions:**

Can you provide more details in experimental setups? Which architectures are used for the target LLMs (15A2B MoE and 1.7B dense models)?

**Ethical Concerns:**

["NO or VERY MINOR ethics concerns only"]

**Limitations:**

Yes

**Quality:**

3

**Strengths And Weaknesses:**

Strengths:
1. This work introduces gating mechanisms within the self-attention layer, accounting for a range of nuances such as positions and granularity.
2. This work conducts comprehensive empirical comparison on both MoE and dense LLMs under various gating mechanisms, investigating which factor may be more impactful in improving the performance of target LLMs. The findings offer insights for readers to understand the significance of different gating settings in architectural design.
3. The gating mechanisms can be used to provide the explanation of gains via non-linearity and input-dependent sparsity.

Weaknesses:
1. The architectures of the target LLMs are limited. It is a challenge to claim whether these findings can be generalized to other architectures such as Llama.
2. This work emphasizes empirical result analysis from a benchmarking perspective, while offering limited investigation into the underlying causes of performance differences across gating configurations. For instance, SDPA output gatings (e.g., elementwise or headwise) appear more effective in mitigating the attention-sink phenomenon. However, it remains unclear why gating applied after SDPA outperforms other variants such as G2, G3, G4, and G5.
3. Table 1, Table 2, and Table 3 are quite confusing in terms of the methods. It seems different gating mechanisms are not compared on the same settings. Which methods (positions like G1, G2, G3, etc are not stated) are compared in Table 2? Is G1 the default setting?

---

> ### Author Rebuttal · Authors · 2025-07-30
>
> > The architectures of the target LLMs are limited. It is a challenge to claim whether these findings can be generalized to other architectures such as Llama.
> >
> - Thank you for the comment. We will include a more detailed description in the revised version. Specifically, our dense model follows the architecture of **Qwen3 1.5B**, and our MoE model adopts the same structure as **Qwen3-30B-A3B**, except that we reduce the number of layers from 48 to 24. More architectural details can be found in the response to your question below.
> - We believe that the Qwen architecture differs only slightly from Llama3, primarily in the design of the FFN and attention components. Since the paper *Massive Activation* has shown that the attention sink phenomenon is widespread across various architectures, and given that our analysis on non-linearity should be applicable to all Transformer-based models, we expect our findings to generalize well.
> - Furthermore, beyond the 28-layer 1.7B model, we have also conducted experiments on a 48-layer 1.7B model with a different width-to-depth ratio, as shown in Table 2.
>
> ---
>
> > This work emphasizes empirical result analysis from a benchmarking perspective, while offering limited investigation into the underlying causes of performance differences across gating configurations. For instance, SDPA output gatings (e.g., elementwise or headwise) appear more effective in mitigating the attention-sink phenomenon. However, it remains unclear why gating applied after SDPA outperforms other variants such as G2, G3, G4, and G5.
> >
> - Thank you for the feedback. Indeed, our current analysis primarily focuses on comparing **G1 (SDPA Elementwise)** and **G2 (v Elementwise)**, explaining why both are effective but G1 performs better: while both enhance non-linearity, the sparsity induced by G2 is limited, and its query-independent gating scores are less effective at mitigating the attention sink. Therefore, G2 underperforms G1 overall.
> - As for the other variants—G3 (k Elementwise), G4 (q Elementwise), and G5 (Dense Output)—they do not effectively introduce non-linearity between the $W_v$ and $W_o$ projections. Our ablation studies further show that gating at these positions induces significantly less sparsity, and the resulting models exhibit similar levels of massive activation and attention sink as the baseline without gating.
> - We will include a detailed analysis of these positions and a discussion on their limited effectiveness in the appendix of the revised manuscript.
>
> ---
>
> > Table 1, Table 2, and Table 3 are quite confusing in terms of the methods. It seems different gating mechanisms are not compared on the same settings. Which methods (positions like G1, G2, G3, etc are not stated) are compared in Table 2? Is G1 the default setting?
> >
> - Thank you for raising this important point. The design rationale behind Tables 1, 2, and 3 is as follows: **Table 1** aims to identify the most effective gating position (which we found to be G1, i.e., after SDPA). Building on this finding, **Table 2** and **Table 3** further evaluate and analyze the impact of applying gating at G1 under different training scales and configurations. Therefore, all gating results in Tables 2 and 3 are based on the **G1 (SDPA output)** position.
> - To avoid confusion, we will clarify this design choice more explicitly in the next version of the paper.
> - Additionally, as mentioned in the response to Reviewer bMKL, we are including the full ablation across gating positions on the 28-layer 1.7B dense model below, which will be added to the appendix:
>
>
>     | Method | Avg PPL | MMLU | GSM8k | Hellaswag | C-eval | CMMLU |
>     | --- | --- | --- | --- | --- | --- | --- |
>     | Baseline | 7.499 | 50.21 | 27.82 | 64.94 | 49.15 | 49.52 |
>     | SDPA Elementwise G1 | 7.404 | 51.15 | 28.28 | 65.48 | 50.72 | 50.72 |
>     | v Elementwise G2 | 7.453 | 50.53 | 27.97 | 65.12 | 49.82 | 49.91 |
>     | k Elementwise G3 | 7.485 | 50.25 | 28.01 | 64.99 | 49.03 | 49.17 |
>     | q Elementwise G4 | 7.466 | 50.33 | 27.89 | 65.04 | 49.15 | 49.33 |
>     | Dense Output G5 | 7.489 | 50.34 | 28.08 | 65.01 | 49.38 | 49.25 |
>
> ---
>
> > Can you provide more details in experimental setups? Which architectures are used for the target LLMs (15A2B MoE and 1.7B dense models)?
> >
> - The detailed architectures for the **1.7B-28 layer**, **1.7B-48 layer**, and **15A2B MoE** models are summarized in the table below:
>
>
>     | Model | 1.7B-28 layer | 1.7B-48 layer | 15A2B MoE |
>     | --- | --- | --- | --- |
>     | Layers | 28 | 48 | 24 |
>     | Query Heads | 16 | 16 | 32 |
>     | Key / Value Heads | 8 | 8 | 4 |
>     | Head Dimension | 128 | 128 | 128 |
>     | Tie Embedding | Yes | Yes | No |
>     | QK Normalization | Yes | Yes | Yes |
>     | Hidden Size | 2048 | 1536 | 2048 |
>     | FFN Size | 6144 | 4608 | 768 |
>     | Number of Experts | – | – | 128 |
>     | Top-k | – | – | 8 |
> - Additionally, we plan to **open-source the 1.7B models** to facilitate further research and reproducibility within the community.

---

### Official Review · Reviewer_Wjax · 2025-06-30

**Clarity:** 4
**Significance:** 4
**Originality:** 3
**Rating:** 6
**Confidence:** 4

**Summary:**

This paper conducts an extensive empirical investigation into incorporating gating mechanisms into the softmax attention module and provides a detailed analysis of the resulting gains and learned patterns. The authors explore over 30 gating variants, including different gating positions (post-q, k, v, Wo. output), granularity (token-wise, head-wise, or head-shared), gating types (additive vs. multiplicative), and activation functions. The study spans both dense models (e.g., 1.7B models trained on 400B or 3.5T tokens) and mixture-of-experts models (e.g., 15A2B trained on 400B tokens), all under a well-optimized training pipeline—covering training data quality, architectural tuning, global batch size, label smoothing, z-loss, and more.

Based on these comprehensive experiments, the paper delivers credible takeaway messages: adding a gating mechanism before the weighted output (Wo) projection in multi-head attention can significantly improve perplexity and performance on a range of downstream benchmarks, including MMLU, GSM8K, and C-Eval.

The paper further attributes these improvements to two key effects: (1) introducing non-linearity in the value-dense low-rank mappings, and (2) enabling sparse, input-dependent modulation that mitigates excessive activations and reduces attention sinks. Detailed ablations support these claims, and the method is shown to improve training stability and generalization to long-context settings up to 128k tokens.

**Questions:**

Table 1 could be further improved by including two additional experiments to strengthen the paper’s findings and conclusions:

1). One of the sub-questions explored in the paper is how the placement of additional parameters affects performance. Currently, the baselines include only GQA settings and expert counts. It would be valuable to add a “more-layer” baseline, especially for the 2.54B activation model. Adding ~200M parameters via extra layers is feasible and would help compare against the current approaches under a similar parameter budget.

2) We also noticed that setting (6) achieves fairly strong performance, despite having only 1/8 the additional parameters of setting (5). This raises the question of whether the performance difference is primarily due to parameter count. Could you consider adding an experiment comparing v-elementwise G2 with multi-head (i.e., n × q × d_k) to control for parameter count and isolate the architectural impact?

**Ethical Concerns:**

["NO or VERY MINOR ethics concerns only"]

**Final Justification:**

After the rebuttal, all my concerns have been addressed, and I have no further questions. I will maintain my strong accept score.

**Quality:**

4

**Strengths And Weaknesses:**

Strengths:

1. The topic addressed in this paper is highly practical and valuable, with strong applicability to structural improvements in large language models (LLMs).
2. The experimental scale and setup are at a production level, lending high credibility and reference value to the conclusions.
3. The takeaway messages are well-reasoned, with thorough analysis and comprehensive experimental support, especially in the ablation and insight sections.

Weaknesses:
N/A

---

> ### Author Rebuttal · Authors · 2025-07-30
>
> > It would be valuable to add a “more-layer” baseline, especially for the 2.54B activation model. Adding ~200M parameters via extra layers is feasible and would help compare against the current approaches under a similar parameter budget.
> >
> - Thank you for this insightful suggestion. We fully agree that a “more-layer” baseline is a strong and valuable architectural comparison, as increasing depth often brings significant performance gains. Following your recommendation, we have conducted additional experiments and will include them in the revised version of the paper:
>
>
>     | Method | Added Total Param (M) | Added Activate Param (M) | Avg PPL | Hellaswag | MMLU | GSM8k | C-eval |
>     | --- | --- | --- | --- | --- | --- | --- | --- |
>     | Baseline | 0 | 0 | 6.026 | 73.07 | 58.79 | 52.92 | 60.26 |
>     | k = 8 | 50 | 50 | 5.979 | 73.51 | 59.78 | 52.16 | 62.26 |
>     | q = 48 | 201 | 201 | 5.953 | 73.59 | 58.45 | 53.30 | 59.67 |
>     | Add 4 Experts | 400 | 0 | 5.964 | 73.19 | 58.84 | 52.54 | **63.19** |
>     | Add More Layer | 625 | 100 | 5.871 | 73.96 | 59.74 | 53.53 | 61.45 |
>     |  |  |  |  |  |  |  |  |
>     | SDPA Elementwise | 201 | 201 | **5.761** | **74.64** | **60.82** | **55.27** | 62.20 |
>
>     For the "Add More Layer" MoE configuration, we separately report both the increase in total parameters and activated parameters, as adding a layer increases both. As shown, adding an extra layer—increasing total parameters by approximately 625M (from 15B)—leads to a noticeable improvement compared to other parameter-increasing methods. However, its performance still falls short of the **SDPA Elementwise** gating approach, which achieves superior results with only 201M additional (and activated) parameters.
>
>
> ---
>
> > We also noticed that setting (6) achieves fairly strong performance, despite having only 1/8 the additional parameters of setting (5). This raises the question of whether the performance difference is primarily due to parameter count. Could you consider adding an experiment comparing v-elementwise G2 with multi-head (i.e., n × q × d_k) to control for parameter count and isolate the architectural impact?
> >
> - Thank you for the suggestion. To better isolate the architectural effect and control for parameter count, we conducted a follow-up experiment using a **Multi-Head Attention (MHA)** variant of the MoE model (changing from q=32, kv=4 to qkv=16). We then compared **G1 (SDPA Elementwise)** and **G2 (v Elementwise)** with *equal parameter budgets* (100M additional parameters each). The results are shown below:
>
>
>     | Method | Added Param (M) | Avg PPL | Hellaswag | MMLU | GSM8k | C-eval |
>     | --- | --- | --- | --- | --- | --- | --- |
>     | Baseline | 0 | 6.085 | 72.84 | 57.68 | 52.01 | 60.41 |
>     | SDPA Elementwise G1 | 100 | **5.842** | **74.01** | **59.54** | **54.43** | 61.34 |
>     | v Elementwise G2 | 100 | 5.900 | 73.32 | 58.52 | 53.84 | **61.51** |
>
>     As observed, when the parameter counts are balanced, the performance gap between G1 and G2 narrows, but **G1 still outperforms G2**. This further supports our hypothesis that the **query-independent sparsity** and **position after SDPA** in G1 contribute meaningfully to its effectiveness, beyond just parameter count.
>
>
> ---
>
> We sincerely appreciate the reviewer’s constructive feedback, which has helped strengthen the paper. We will incorporate these experiments and discussions into the final version.

---

### Official Review · Reviewer_bMKL · 2025-07-03

**Clarity:** 3
**Significance:** 2
**Originality:** 3
**Rating:** 4
**Confidence:** 4

**Summary:**

The paper explores the effects of gating at various positions of self-attention, i.e., after query/key/value projections, after the self-attention output, and after the final dense layer.  Two different model types (MoE, Dense) with architectural/training recipe variations are trained with the aforementioned gating placements.  The paper reports qualitative results for both MoE and Dense models, showing SDPA output gating effectively improves performance across standard natural tasks.  The source of this improvement is thus explored, showing that gating location can greatly affect the sparsity of attention scores and, subsequently, this can mitigate massive activations and attention sinks during training.

**Questions:**

-Why were the same placement experiments in Table 1 not repeated for the dense model in Table 2?  The missing "Max LR" experiments for different configurations (e.g., (9) SDPA Headwise trained with max LR 8.0 × 10−3) make it difficult to draw definitive conclusions.

## Incorrect citatios
> Prior work has established that while increased network depth, large learning rates, and large batch sizes can significantly improve
model performance [33, 34, 35] and distributed training efficiency, they often introduce training instabilities [34, 36, 37].

Citations 34 and 36 do not support the above claim.  34 attributed instability to pre vs post residual placement of the norm, while 36 did not attribute instability to any particular factor (but rather explored layernorm placement's effect on stability).

**Ethical Concerns:**

["NO or VERY MINOR ethics concerns only"]

**Final Justification:**

# Post rebuttal update

The authors have added additional experiments and discussion during the rebuttal to address my primary concerns regarding disconnects in the original experiments and the claims from the paper.  Indeed, the main takeaway regarding stability under gating looks sound and I've increased my score accordingly.

**Limitations:**

Yes

**Quality:**

2

**Strengths And Weaknesses:**

# Strengths
## Clarity, Significance, and Originality
The paper is well written and easy to understand.  The topic is interesting; a nuanced, controlled study of the role gating plays in Transformer models can have a large impact.  In terms of originality, the specific study has the potential to separate itself from previous works in the area.

## Quality
The experiments make sense and are in line with the motivations of the paper.  The explainability results in Section 4 were also good contributions to explain the effects of gating-placement has on attention scores.  Although it is not surprising that SDPA Elementwise Gate induces the most sparsity, it is nice to see this verified.

# Weaknesses
## Quality
Although the paper seeks to assess gating-location's contribution without confounding factors, key takeaways are made without sufficient evidence.  In particular,
> Applying this method to dense transformers further demonstrates that the gate enables stable training with larger batch sizes and learning rates, resulting in improved performance.

It is difficult to make this claim definitively, since the batch size is varied with the total training tokens.  Thus, a carefully isolated experiment would compare: a) 28 Layer, 1.7B Parameters, 400B Tokens, Batch Size=1024, (b) 28 Layer, 1.7B Parameters, 3.5T Tokens, Batch Size=1024, (c) 28 Layer, 1.7B Parameters, 400B Tokens, Batch Size=2048, (d) 28 Layer, 1.7B Parameters, 3.5T Tokens, Batch Size=2048.  Instead, the only comparison we have is between (a) and (d).  This is the same for the 48 layers experiments.

## Significance
Some of the performance improvements are not as substantial as reported.  For example, overall, results in Table 1 are not significant performance improvements.  It is difficult to say "significant reduction in PPL" in Table 3.  Similarly, it is difficult to call a 0.2 PPL reduction a significant performance improvement.  For dense models, while gating-placement was much more impactful on 48 layer 1T pretraining token models, gains are modest for the 28 layer model.  It is possible that substantial improvements may have occurred for the missing experiments above.

## Originality
> Despite its widespread adoption and empirical success, the function and impact of gating mechanisms remain insufficiently explored beyond their initial intuition.

This claim is too broad, the impact of gating has been explored in linear attention [1] and standard attention [2,3] networks.

# References
[1] Chen, Yingfa, et al. "Stuffed Mamba: Oversized States Lead to the Inability to Forget." arXiv preprint arXiv:2410.07145 (2024).

[2] Lin, Zhixuan, et al. "Forgetting transformer: Softmax attention with a forget gate." arXiv preprint arXiv:2503.02130 (2025).

[3] Xue, Lanqing, Xiaopeng Li, and Nevin L. Zhang. "Not all attention is needed: Gated attention network for sequence data." Proceedings of the AAAI conference on artificial intelligence. Vol. 34. No. 04. 2020.

---

> ### Author Rebuttal · Authors · 2025-07-30
>
> ## Weaknesses
> ### Quality
> > Applying this method to dense transformers further demonstrates that the gate enables stable training with larger batch sizes and learning rates, resulting in improved performance.` It is difficult to make this claim definitively...
>   - Thank you for this suggestion. We increased the batch size when increasing the number of training tokens because we aimed to compare under the best possible baseline performance. Our hyperparameter search for the baseline revealed that both batch size and learning rate need to be increased as the number of training tokens grows, which is consistent with existing scaling laws (`Predictable Scale: Part I, Step Law -- Optimal Hyperparameter Scaling Law in Large Language Model Pretraining`). We understand that this setup may introduce additional confounding factors (i.e., token count) when comparing the effect of gating under different batch sizes. Therefore, we have further added experiments under setting (c):
>
>
>     | Method | LR | Tokens | Bsz | Avg PPL | MMLU | HumanEval | GSM8k | Hellaswag | C-eval | CMMLU |
>     | --- | --- | --- | --- | --- | --- | --- | --- | --- | --- | --- |
>     | Baseline | 4e-3 | 400B | 1024 | 7.499 | 50.21 | 28.66 | 27.82 | 64.94 | 49.15 | 49.52 |
>     | SDPA Elementwise | 4e-3 | 400B | 1024 | **7.404** | **51.15** | **29.27** | **28.28** | **65.48** | **50.72** | **50.72** |
>     |  |  |  |  |  |  |  |  |  |  |  |
>     | Baseline | 4e-3 | 400B | 2048 | 7.538 | 49.02 | 24.37 | 25.72 | 63.54 | 48.02 | 48.82 |
>     | SDPA Elementwise | 4e-3 | 400B | 2048 | 7.487 | 50.41 | 27.56 | 27.13 | 64.11 | 49.64 | 50.03 |
>   - We observe that increasing the batch size larger than the optimal one leads to a performance drop. However, adding the SDPA output gate still results in better performance than the baseline, with more pronounced improvements on benchmarks such as MMLU, HumanEval, and GSM8k.
>   - Nevertheless, we acknowledge that the phrase `resulting in improved performance` could be misleading and that the comparison remains challenging. In the next version, we will revise this statement to `also resulting in performance improvement relative to the baseline.`
>
> ---
>
> ### Significance
>
> > For example, overall, results in Table 1 are not significant performance improvements. It is difficult to say "significant reduction in PPL" in Table 3. Similarly, it is difficult to call a 0.2 PPL reduction a significant performance improvement...
>
> - We would like to provide additional context to demonstrate that a 0.2 PPL reduction is a meaningful performance gain:
>     - In our Table 2, for the 48-layer model, increasing training tokens from 400B to 1T leads to only a 0.06 reduction in PPL for the baseline (from 7.4 to 7.34), while the SDPA Elementwise gate achieves a 0.187 reduction (from 7.28 to 7.093). In contrast, in Table 1, the gate reduces PPL from ~6.0 to ~5.8—a 0.2 drop—even though architectures differ, this remains a substantial improvement.
>     - Additionally, we supplement Table 1’s setting with baseline models trained for 500B and 600B tokens (with full cosine schedulers), as shown below:
>
>
>         | Method | Tokens | Avg PPL | MMLU | GSM8k |
>         | --- | --- | --- | --- | --- |
>         | Baseline | 400B | 6.026 | 58.79 | 52.92 |
>         | Baseline | 500B | 5.851 | 60.17 | 54.85 |
>         | Baseline | 600B | **5.704** | **61.03** | 55.15 |
>         |  |  |  |  |  |
>         | SDPA Elementwise | 400B | 5.761 | 60.82 | **55.27** |
>     - Notably, the SDPA Elementwise model trained on 400B tokens outperforms the baseline trained on 500B tokens and is only slightly behind the 600B baseline.
>     - For further reference, in the Fox paper (mentioned by the reviewer), Table 3 shows that adding an output gate in a 360M model trained on 7.5B tokens reduces PPL from 7.08 to 6.88—a 0.2 drop. While test PPLs are not directly comparable due to different datasets, this supports our claim that a 0.2 PPL reduction is meaningful.
>     - Moreover, we will remove the word `significant` to avoid mis-understandings.
>
> ---
>
> > For dense models, while gating-placement was much more impactful on 48-layer 1T pretraining token models, gains are modest for the 28-layer model. It is possible that substantial improvements may have occurred for the missing experiments above.
> >
> - Thank you for raising this point. We have further supplemented experiments on the 28-layer 1.7B model trained on 400B tokens with increased learning rates. The results are shown in the table below:
>
>     | Method | LR | Avg PPL | HumanEval | MMLU | GSM8k | Hellaswag | C-eval | CMMLU |
>     | --- | --- | --- | --- | --- | --- | --- | --- | --- |
>     | Baseline | 4e-3 | 7.499 | 28.66 | 50.21 | 27.82 | 64.94 | 49.15 | 49.52 |
>     | SDPA Elementwise | 4e-3 | **7.404** | 29.27 | 51.15 | 28.28 | **65.48** | 50.72 | 50.72 |
>     | Baseline | 8e-3 | 8.941 | 20.13 | 41.17 | 16.15 | 58.10 | 44.36 | 45.13 |
>     | SDPA Elementwise | 8e-3 | 7.425 | **30.37** | **52.51** | **29.24** | 65.30 | **51.76** | **51.88** |
>
>     As shown, when the learning rate is further increased, the baseline model encounters convergence issues and suffers significant performance degradation. In contrast, the model with SDPA Elementwise gating maintains stable training and achieves further improvements across all benchmarks except PPL and Hellaswag. This supports our earlier claim that the gating mechanism enables stable and effective training under larger learning rates.
> ---
> ### Originality
>
> > "Despite its widespread adoption and empirical success, the function and impact of gating mechanisms remain insufficiently explored beyond their initial intuition."
> > This claim is too broad, the impact of gating has been explored in linear attention [1] and standard attention [2,3] networks.
> >
> - Thank you for the feedback. We have discussed the relation to [2] in line 300. In the revised version, we will add the following clarifications regarding [1] and [3]:
>     - [1] discusses the difficulty of Mamba’s decay mechanism in forgetting historical hidden states and also uses an output gate at the SSM layer. However, neither [1] nor the original Mamba2 paper analyzes the behavior or impact of the output gate in detail.
>     - [3] proposes a Gumbel-Softmax-based gate in LSTM attention to mask irrelevant tokens for the current query. In contrast to [3], which applies a hard mask *before* SDPA to explicitly remove irrelevant token information, our attention output gate operates *after* SDPA and uses gating scores to suppress irrelevant information, effectively mitigating the attention sink phenomenon.
>     - We will modify the expression to `Despite its widespread adoption and empirical success, most recent works do not look into the gating mechanisms like the gating scores and their effect on the model’s hidden states.`
>
> ---
> ## Questions:
>
> > Why were the same placement experiments in Table 1 not repeated for the dense model in Table 2? The missing "Max LR" experiments for different configurations (e.g., (9) SDPA Headwise trained with max LR 8.0 × 10⁻³) make it difficult to draw definitive conclusions.
> >
> - Initially, we focused on evaluating gating positions in MoE models due to their lower training cost and strong performance, expecting similar trends in dense models. We now supplement the results for different gating placements in a 28-layer, 1.7B dense model trained on 400B tokens. These will be added to the appendix:
>
>
>     | Method | Avg PPL | MMLU | GSM8k | Hellaswag | C-eval | CMMLU |
>     | --- | --- | --- | --- | --- | --- | --- |
>     | Baseline | 7.499 | 50.21 | 27.82 | 64.94 | 49.15 | 49.52 |
>     | SDPA Elementwise G1 | **7.404** | **51.15** | **28.28** | **65.48** | **50.72** | **50.72** |
>     | v Elementwise G2 | 7.453 | 50.53 | 27.97 | 65.12 | 49.82 | 49.91 |
>     | k Elementwise G3 | 7.485 | 50.25 | 28.01 | 64.99 | 49.03 | 49.17 |
>     | q Elementwise G4 | 7.466 | 50.33 | 27.89 | 65.04 | 49.15 | 49.33 |
>     | Dense Output G5 | 7.489 | 50.34 | 28.08 | 65.01 | 49.38 | 49.25 |
> - We further supplement results for `SDPA Headwise` at LR = 8e-3:
>
>
>     | Method | Avg PPL | MMLU | GSM8k | Hellaswag | C-eval | CMMLU |
>     | --- | --- | --- | --- | --- | --- | --- |
>     | SDPA Elementwise | **7.325** | **54.47** | **36.62** | **66.40** | 53.91 | **53.80** |
>     | SDPA Headwise | 7.384 | 54.13 | 34.56 | 65.97 | **54.10** | 52.38 |
> - We observe that under more aggressive training setting, the elementwise gate outperforms the headwise variant, and the training curve is smoother. Hence, we recommend the elementwise gating design.
>
> ---
> ### Incorrect citations
>
> > "Prior work has established that while increased network depth, large learning rates, and large batch sizes can significantly improve model performance [33, 34, 35] and distributed training efficiency, they often introduce training instabilities [34, 36, 37]."
> >
> >
> > Citations 34 and 36 do not support the above claim. 34 attributed instability to pre vs post residual placement of the norm, while 36 did not attribute instability to any particular factor (but rather explored layernorm placement's effect on stability).
> >
> - Thank you for pointing this out. The confusion likely arose from combining multiple factors across citations.
>     - For [34], we intended to use its Table 1 to support the claim that increased depth can cause instability (many models diverge when depth increases), and its Table 2 to support that depth can improve performance (both baseline and DeepNet show gains with more layers).
>     - For [36], a more appropriate citation would be `A Theory on Adam Instability in Large-Scale Machine Learning`, which thoroughly discusses how larger learning rates and batch sizes can lead to loss spikes and other instabilities. [36] primarily proposes techniques like Embedding Layer Gradient Shrink (EGS) to address such spikes .
>     - We will modify the expression accordingly.
>
> Thanks again for your suggestions!

---

> > ### Author Response · Authors · 2025-08-04
> >
> > Dear Reviewer bMKL,
> >
> > We sincerely appreciate your constructive review. As the discussion period is expected to conclude in the next few days, please let us know if you have any additional concerns regarding our work. We are happy to address any further questions. However, if you are satisfied with our responses, we kindly hope you will consider adjusting your rating accordingly.
> >
> > Thank you,
> >
> > The Authors

---

> > ### Comment · Reviewer_bMKL · 2025-08-05
> > **Reply**
> >
> > Thanks to the authors for their response and for the additional experiments.
> >
> > > For further reference, in the Fox paper (mentioned by the reviewer), Table 3 shows that adding an output gate in a 360M model trained on 7.5B tokens reduces PPL from 7.08 to 6.88—a 0.2 drop. While test PPLs are not directly comparable due to different datasets, this supports our claim that a 0.2 PPL reduction is meaningful.
> >
> > A 0.2 PPL drop for a 360M model trained on 7.5 B tokens is a far different scenario than the model sizes and token counts considered in the paper.
> >
> > > In our Table 2, for the 48-layer model, increasing training tokens from 400B to 1T leads to only a 0.06 reduction in PPL for the baseline (from 7.4 to 7.34), while the SDPA Elementwise gate achieves a 0.187 reduction (from 7.28 to 7.093). In contrast, in Table 1, the gate reduces PPL from ~6.0 to ~5.8—a 0.2 drop—even though architectures differ, this remains a substantial improvement.
> >
> > Thank you for the explanation, could the authors please include this discussion in the paper?  This better conveys the significance of the reduction in PPL.
> >
> > The described Originality and Incorrect Citations are also appreciated, I am looking forward to these changes in the paper.
> >
> > > Thank you for raising this point. We have further supplemented experiments on the 28-layer 1.7B model trained on 400B tokens with increased learning rates. The results are shown in the table below:
> >
> > Thank you, I agree these new results support the original claim.
> >
> > > We observe that increasing the batch size larger than the optimal one leads to a performance drop. However, adding the SDPA output gate still results in better performance than the baseline, with more pronounced improvements on benchmarks such as MMLU, HumanEval, and GSM8k.
> >
> > An optimal batch size and the above result lie at odds with the original claim:
> > > Applying this method to dense transformers further demonstrates that the gate enables stable training with larger batch sizes and learning rates
> >
> > Could the authors revise this portion of the claim?

---

> ### Author Response · Authors · 2025-08-05
> **Response to Reviewer bMKL**
>
> Dear Reviewer bMKL,
>
> Thank you very much for your thoughtful follow-up and for acknowledging our additional experiments. We are glad that the new results on larger learning rates support the claim regarding training stability, and we appreciate your suggestion to include the discussion on PPL reduction in the revised manuscript.
>
> **As we can't update the PDF now**, we will incorporate this context into the paper to better convey the significance of the observed improvements.
>
> We also sincerely thank you for pointing out the potential ambiguity in our original statement:
>
> > *"Applying this method to dense transformers further demonstrates that the gate enables stable training with larger batch sizes and learning rates."*
>
> We agree that this phrasing could be misinterpreted, and we appreciate the opportunity to clarify our intended meaning.
>
> Our key observation is that **the SDPA output gate improves training stability**, allowing the model to converge successfully under larger learning rates and larger batch sizes—settings that often lead to instability (e.g., loss spikes, divergence) in baseline models. This does *not* imply that increasing batch size or learning rate alone necessarily leads to better final performance; indeed, as shown in our new experiments, increasing the batch size beyond the optimal point (without adjusting other hyperparameters) can degrade performance due to fewer update steps or suboptimal optimization dynamics.
>
> However, with the gating mechanism, the model remains **stable and trainable** in these more aggressive regimes. For instance, at a high learning rate of $8\times10^{-3}$, the baseline diverges or degrades severely, while the gated model maintains stable convergence and achieves better performance across multiple benchmarks. This suggests that the gate expands the effective hyperparameter landscape in which stable training is possible.
>
> Moreover, the optimal hyperparameter configuration may shift when introducing the gate. Our experiments were designed to first identify the best baseline configuration (e.g., bsz=1024, LR=4e-3 for 400B tokens), then evaluate the gated variant under the same or more aggressive settings—without re-optimizing all hyperparameters. The consistent gains observed under these conditions demonstrate the robustness and practical utility of the gating mechanism.
>
> To avoid misinterpretation, we will revise the claim in the final version as follows:
>
> > *"Incorporating the SDPA output gate enables stable training under larger learning rates and batch sizes—regimes where the baseline often becomes unstable. This suggests that the optimal hyperparameter configuration shifts when using gating. In practice, one effective way to leverage the gate is to start from the baseline’s optimal batch size and moderately increase the learning rate. Further jointly tuning batch size and learning rate may yield additional gains."*
>
> We believe this more precise wording reflects both the empirical findings and the underlying intent of our analysis.
>
> Thank you again for your insightful feedback, which has helped us significantly improve the clarity and rigor of our work. We are grateful for your time and engagement throughout the review process.
>
> Best regards,
> The Authors

---

> > ### Author Response · Authors · 2025-08-07
> >
> > Dear Reviewer bMKL,
> >
> > Thank you very much for your thoughtful and technically insightful review, and for taking the time to engage with our responses and the additional experiments we provided.
> >
> > As the discussion period is drawing to a close, we would like to kindly ask if there are any remaining points from your review or our rebuttal where you feel further clarification would be beneficial? We are committed to ensuring our work is as clear and robust as possible and are happy to provide any additional explanation or data if needed.
> >
> > Thank you again for your valuable feedback and your careful consideration of our work.
> >
> > Best regards,
> >
> > The Authors

---

### Official Review · Reviewer_cPo5 · 2025-07-03

**Clarity:** 4
**Significance:** 3
**Originality:** 3
**Rating:** 6
**Confidence:** 5

**Summary:**

The paper studies the effect of introducing gating at different stages of multi-head attention. The strongest point of the paper is the comprehensive study and comparison of various options. The authors find that applying gating after the value matrix or after head concatenation provides most of the benefits, with gains of up to 2% on MMLU. Additionally, the paper demonstrates that such gating reduces the effect of attention sink and massive activations, leading to easier long-context finetuning.

**Questions:**

It will be great if authors can comment on weaknesses above.

**Ethical Concerns:**

["NO or VERY MINOR ethics concerns only"]

**Limitations:**

Mentioned

**Quality:**

4

**Strengths And Weaknesses:**

**Strengths**
1) Proposed analysis is novel and makes a lot of sense.
2) The depth of ablations is impressive: position, granularity, head-specific or shared, multiplicative or additive, activation functions — all are studied.
3) MoEs and dense models are considered, training scale is reasonable.
4) Long context, sparsity, and attention sink are studied.
5) Each experiment is followed by a heat summary; it is quite enjoyable to read and learn as you go.

**Weaknesses**
1) No major weaknesses, only a couple of suggestions.
2) One suggestion is to summarize a single learning and a single recommendation on the best way to apply gating. Currently, multiple options are highlighted, but narrowing down to one can be beneficial for readers.
3) The attention sink can be suppressed by using "Quiet Attention" or Meta tokens from [R1]. It would be great to see if such techniques are complementary.
4) The reduction of massive activations should be beneficial for quantization. Adding post-training quantization results would be beneficial.

[R1] Dong, Xin, Yonggan Fu, Shizhe Diao, Wonmin Byeon, Zijia Chen, Ameya Sunil Mahabaleshwarkar, Shih-Yang Liu et al. "Hymba: A hybrid-head architecture for small language models." arXiv preprint arXiv:2411.13676 (2024).

---

> ### Author Rebuttal · Authors · 2025-07-30
>
> > One suggestion is to summarize a single learning and a single recommendation on the best way to apply gating. Currently, multiple options are highlighted, but narrowing down to one can be beneficial for readers.
> >
> - Thank you for the suggestion! Our top recommendation is to apply **elementwise gating at the SDPA output** (i.e., after the attention-weighted value projection) in combination with a moderately increased learning rate.
> - Furthermore, our follow-up experiments reveal that increasing the head dimension (e.g., changing from q=32, kv=4, head_dim=128 to q=16, kv=2, head_dim=256) while keeping the total model size constant can further improve performance. However, such a configuration leads to more frequent training instabilities (e.g., loss spikes) in the baseline. In contrast, with **elementwise SDPA gating**, the model remains stable during training.
> - Therefore, more generally, we recommend using **elementwise SDPA output gating** as a foundational improvement, upon which more aggressive training strategies (e.g., larger learning rates, larger head dimensions) can be safely combined.
>
> ---
>
> > The attention sink can be suppressed by using "Quiet Attention" or Meta tokens from [R1]. It would be great to see if such techniques are complementary.
> >
> - Thank you for the insightful suggestion. We will include a discussion on **Meta tokens** in the appendix. In our view, Meta tokens do not truly *resolve* the attention sink issue but rather *redirect* it to a designated special token. Once the model learns to suppress the sink phenomenon via our gating mechanism, the role of the Meta token may be reduced to that of a soft prompt, providing only marginal additional gains.
>
> ---
>
> > The reduction of massive activations should be beneficial for quantization. Adding post-training quantization results would be beneficial.
> >
> - Thank you for the comment. While we currently do not include quantization results in the paper, we plan to release **quantized versions** of our models alongside the full-precision checkpoints to facilitate community research in this direction.

---

### Public Comment · ~Linfan_Zhang1 · 2026-01-19
**Additive is better?**

Hi, in table 1, row (14) uses additive gating and activation function SiLU. To conclude that additive gating is better, should I compare with row (5) or (15)? Comparing with row (5), then we would have two component changes, so it is unclear which one contributes to the improvement. Comparing with row (15), then some metrics in row (14) are actually better. I wonder if there is a typo in the activation function for row (14)?

---

### Note · Authors · 2025-08-12

We sincerely thank the reviewers and Area Chairs for their insightful, constructive, and thorough feedback throughout the review process. We have carefully addressed all concerns raised by the reviewers.

Our work presents a comprehensive empirical study on gating mechanisms within softmax attention, demonstrating that applying an elementwise gate at the SDPA output yields significant improvements in model performance, training stability, and long-context generalization—particularly under aggressive training regimes (e.g., higher learning rates, larger batch sizes). The depth of our ablation studies—including position, granularity, activation functions, and architectural variants across both MoE and dense models—provides robust evidence for this finding.

In response to reviewer feedback, we have:

- Added new experiments to isolate the effects of batch size and learning rate, confirming that the gate enables stable training under more aggressive settings where baselines fail.
- Provided additional context to clarify the significance of performance gains, including comparisons to extended baselines and prior work (e.g., PPL drops of ~0.2 are meaningful in context).
- Clarified experimental design and table interpretations, explicitly stating that Tables 2 and 3 build upon the G1 (SDPA Elementwise) configuration identified as optimal in Table 1.
- Expanded analysis on why gating after SDPA outperforms other placements, attributing it to enhanced non-linearity and input-dependent sparsity that effectively mitigate attention sink and massive activations.
- Refined citations and claims regarding training stability and originality.
- Will include detailed architectural specifications, full ablation results on dense models, and discussions on quantization and Meta tokens in the final manuscript.

We appreciate the opportunity to engage with the community and look forward to the final decision.

---

### Decision · Program_Chairs · 2025-09-17

**Decision:**

Accept (oral)

**Comment:**

In this work, the authors do a study of gating mechanisms which tend to be almost omnipresent nowadays, especially in the context of LLMs. To do so they introduce a gating in self-attention layers. However, they go way beyond that and study a different range of gating mechanisms (over thirty of them) on where to put the gating, the different granularities of it providing a very complete and comprehensive study. Their change in the mechanism of softmax not only improve the results in different benchmarks, but also show a reduction in both activations and softmax weight in particular tokens (studied before in different terms as 'Attention-Sink' or 'Registers'), which can have other practical advantages such as simplifying the quantization.

The paper initially had some diverging reviews, with 2 strong accepts, 1 accept and 1 borderline reject. Reviewer cPo5 who is an authority on the field found no weaknesses at all with the paper, having only a couple of suggestions which the authors addressed. Similarity, reviewer Wjax finds no weaknesses at all, and like Reviewer cPo5 scores the paper with the highest score. Reviewer pXLA asks for experiments in modern LLMs, and for the detailed hyperparameters. The authors response that they will add experiments in QWen architecture, give the hyperparameters, and promise to open-source one of the models.

Reviewer bMKL is the only reviewer to initially score the paper in the negative region (Borderline reject). They have some doubts on the experimental section. In particular, they are not convinced on the results because of the varying batch size, to which the authors respond in showing experiments with an increased batch size when increasing the number of training tokens, similar to what was the reviewer's suggestion. The reviewer also is not convinced that a 0.2 improvement in PPL is not significant. The authors and the AC disagree on this point with the reviewer, while the other reviewers found the improvement significant. Finally, the reviewer had some smaller points which were properly addressed by the authors. During the authors-reviewer phase, the reviewer initially engaged with the authors, and after the stage, they increased the score to 4 (Borderline Accept) claiming that the authors 'added additional experiments and discussion during the rebuttal to address my primary concerns'. At this stage, there is a high consensus between the reviewers for the paper to be accepted with scores 6, 6, 5, 4.

Having read the paper and the reviews, I tend to agree with the reviewer's comments. This is a high-quality paper that deserves to be accepted to NeurIPS. In particular:

- This is a very-well executed study with an impressive number of experiments, and as Reviewer Wjax it is indeed at a production level.

- The paper is well-written, enjoyable to read with arguments being supported by tons of evidence.

- Strong empirical results in a different range of architectures, models and token size.

- Important findings.

I think this paper will be received well by the community, and will improve the knowledge on the gating mechanism in LLMs, if not improving the quantization schemes. I would go as far as claiming that this has been one of the top 5 papers I have read this year. The authors are urged to add the parts they mentioned in the rebuttal for the camera-ready version of the paper. Congratulations to the authors for a perfectly executed study!